# The Scale Invariant Vacuum Paradigm: Main Results and Current Progress

Vesselin G. Gueorguiev [1,2,*,†] and Andre Maeder [3]

1  Institute for Advanced Physical Studies, 1784 Sofia, Bulgaria
2  Ronin Institute for Independent Scholarship, 127 Haddon Pl., Montclair, NJ 07043, USA
3  Geneva Observatory, University of Geneva, Chemin des Maillettes 51, CH-1290 Sauverny, Switzerland; andre.maeder@unige.ch
*  Correspondence: vesselin@mailaps.org
†  This paper is a summary of the talk presented at the conference: Alternative Gravities and Fundamental Cosmology (AlteCosmoFun'21), Organized by the University of Szczecin, Szczecin, Poland, 6–10 September 2021.

**Abstract:** We present a summary of the main results within the Scale Invariant Vacuum (SIV) paradigm as related to the Weyl Integrable Geometry (WIG) as an extension to the standard Einstein General Relativity (EGR). After a brief review of the mathematical framework, we will highlight the main results related to inflation within the SIV, the growth of the density fluctuations, and the application of the SIV to scale-invariant dynamics of galaxies, MOND, dark matter, and the dwarf spheroidals. The possible connection between the weak-field SIV equations and the notion of un-proper time parametrization within the reparametrization paradigm is also discussed.

**Keywords:** cosmology: theory, dark matter, dark energy, inflation; galaxies: formation, rotation; Weyl integrable geometry; Dirac co-calculus

## 1. Motivation

The paper is a summary of the current main results within the Scale Invariant Vacuum (SIV) paradigm as related to the Weyl Integrable Geometry (WIG) as an extension to the standard Einstein General Relativity (EGR). It is a reflection of the corresponding presentation at the conference: Alternative Gravities and Fundamental Cosmology (AlteCosmoFun'21), organized by the University of Szczecin, Poland, 6–10 September 2021.

After a general introduction on the problem of scale invariance and physical reality, along with the similarities and differences of Einstein General Relativity and Weyl Integrable Geometry, we briefly review the mathematical framework as pertained to Weyl Integrable Geometry, Dirac Co-Calculus, and reparametrization invariance. Rather than re-deriving the weak-field SIV results for the equations of motion, we have decided to use the idea of reparametrization invariance [1] to illustrate the corresponding equations of motion. The relevant discussion on reparametrization invariance is in the section on the Consequences of Going beyond Einstein's General Relativity. This section precedes the brief review of the necessary results about the Scale Invariant Cosmology idea needed in the section on Comparisons and Applications, where we highlight the main results related to inflation within the SIV [2], the growth of the density fluctuations [3], and the application of the SIV to scale-invariant dynamics of galaxies, MOND, dark matter, and the dwarf spheroidals [4]. We end the paper with a section containing the Conclusions and Outlook for future research directions.

### 1.1. Scale Invariance and Physical Reality

The presence of a scale is related to the existence of physical connection and causality. The corresponding relationships are formulated as physical laws dressed in mathematical

expressions. The laws of physics (numerical factors in the formulae) change upon change of scale, but maintain a form-invariance. As a result, using consistent units is paramount in physics and leads to powerful dimensional estimates of the order of magnitude of physical quantities based on a simple dimensional analysis. The underlined scale is closely related to the presence of material content, which reflects the energy scale involved.

However, in the absence of matter, a scale is not easy to define. Therefore, an empty universe would be expected to be scale invariant! Absence of scale is confirmed by the scale invariance of the Maxwell equations in vacuum (no charges and no currents—the sources of the electromagnetic fields). The field equations of general relativity are scale invariant for empty space with zero cosmological constant. What amount of matter is sufficient to kill scale invariance is still an open question. Such a question is particularly relevant to cosmology and the evolution of the universe.

### 1.2. Einstein General Relativity (EGR) and Weyl Integrable Geometry (WIG)

Einstein's General Relativity (EGR) is based on the premise of a torsion-free covariant connection that is metric-compatible and guarantees the preservation of the length of vectors along geodesics $(\delta \| \overrightarrow{v} \| = 0)$. The theory has been successfully tested at various scales, starting from local Earth laboratories, the Solar system, on galactic scales via light-bending effects, and even on an extragalactic level via the observation of gravitational waves. The EGR is also the foundation for modern cosmology and astrophysics. However, at galactic and cosmic scales, some new and mysterious phenomena have appeared. The explanations for these phenomena are often attributed to unknown matter particles or fields that are yet to be detected in our laboratories—dark matter and dark energy.

As no new particles or fields have been detected in the Earth labs for more than twenty years, it seems reasonable to revisit some old ideas that have been proposed as a modification of EGR. In 1918, Weyl proposed and extension by adding local gauge (scale) invariance [5]. Other approaches were more radical by adding extra dimensions, such as Kaluza–Klein unification theory. Then, via Jordan conformal equivalence, one comes back to the usual 4D spacetime as projective relativity theory, but with at least one additional scalar field. Such theories are also known as Jordan–Brans–Dicke scalar-tensor gravitation theories [6,7]. In most such theories, there is a major drawback—a varying Newton constant $G$. As no such variations have been observed, we prefer to view Newton's gravitational constant $G$ as constant despite the experimental issues on its measurements [8].

In the light of the above discussion one may naturally ask: could the mysterious phenomena be artifacts of non-zero $\delta \| \overrightarrow{v} \|$, but often negligible and with almost zero value $(\delta \| \overrightarrow{v} \| \approx 0)$, which could accumulate over cosmic distances and fool us that the observed phenomena may be due to dark matter and/or dark energy? An idea of extension of EGR was proposed by Weyl as soon as the General Relativity (GR) was proposed by Einstein. Weyl proposed an extension to GR by adding local gauge (scale) invariance that does have the consequence that lengths may not be preserved upon parallel transport. However, it was quickly argued that such a model will result in a path dependent phenomenon and, thus, contradicting observations. A remedy was later found to this objection by introducing Weyl Integrable Geometry (WIG), where the lengths of vectors are conserved only along closed paths $(\oint \delta \| \overrightarrow{v} \| = 0)$. Such formulation of the Weyl's original idea defeats the Einstein objection! Furthermore, given that all we observe about the distant universe are waves that reach us, the condition for Weyl Integrable Geometry is basically saying that the information that arrives to us via different paths is interfering constructively to build a consistent picture of the source object.

One way to build a WIG model is to consider conformal transformation of the metric field $g'_{\mu\nu} = \lambda^2 g_{\mu\nu}$ and to apply it to various observational phenomena. As we will see in the discussion below, the demand for homogeneous and isotropic space restricts the field $\lambda$ to depend only on the cosmic time and not on the space coordinates. The weak field limit of such a WIG model results in an extra acceleration in the equation of motion that is proportional to the velocity of the particle.

This behavior is somewhat similar to the Jordan–Brans–Dicke scalar-tensor gravitation; however, the conformal factor $\lambda$ does not seems to be a typical scalar field as in the Jordan–Brans–Dicke theory [6,7].

The Scale Invariant Vacuum (SIV) idea provides a way of finding out the specific functional form of $\lambda(t)$ as applicable to LFRW cosmology and its WIG extension.

We also find it important to point out that extra acceleration in the equations of motion, which is proportional to the velocity of a particle, could also be justified by requiring re-parametrization symmetry. Not implementing re-parametrization invariance in a model could lead to un-proper time parametrization [1] that seems to induce "fictitious forces" in the equations of motion similar to the forces derived in the weak field SIV regime. It is a puzzling observation that may help us understand nature better.

## 2. Mathematical Framework

The framework for the Scale Invariant Vacuum paradigm is based on the Weyl Integrable Geometry and Dirac co-calculus as mathematical tools for description of nature [5,9].

### 2.1. Weyl Integrable Geometry and Dirac Co-Calculus

The original Weyl Geometry uses a metric tensor field $g_{\mu\nu}$, along with a "connexion" vector field $\kappa_\mu$, and a scalar field $\lambda$. In the Weyl Integrable Geometry, the "connexion" vector field $\kappa_\mu$ is not an independent field, but it is derivable from the scalar field $\lambda$.

$$\kappa_\mu = -\partial_\mu \ln(\lambda) \tag{1}$$

This form of the "connexion" vector field $\kappa_\mu$ guarantees its irrelevance, in the covariant derivatives, upon integration over closed paths. That is, $\oint \kappa_\mu dx^\mu = 0$. In other words, $\kappa_\mu dx^\mu$ represents a closed 1-form; furthermore, it is an exact form, as (1) implies $\kappa_\mu dx^\mu = d\lambda$. Thus, the scalar function $\lambda$ plays a key role in the Weyl Integrable Geometry. Its physical meaning is related to the freedom of a local scale gauge that provides a description upon change in scale via local re-scaling $l' \to \lambda(x)l$.

#### 2.1.1. Gauge Change and Derivatives within the EGR and WIG Context

The covariant derivatives use the rules of the Dirac co-calculus [9] where tensors also have co-tensor powers based on the way they transform upon change of scale. For the metric tensor $g_{\mu\nu}$ this power is $n = 2$. This follows from the way the length of a line segment $ds$ with coordinates $dx^\mu$ is defined via the usual expression $ds^2 = g_{\mu\nu}dx^\mu dx^\nu$.

$$l' \to \lambda(x)l \Leftrightarrow ds' = \lambda ds \Rightarrow g'_{\mu\nu} = \lambda^2 g_{\mu\nu}$$

This leads to $g^{\mu\nu}$ having the co-tensor power of $n = -2$ in order to have the Kronecker $\delta$ as scale invariant object ($g_{\mu\nu}g^{\nu\rho} = \delta^\rho_\mu$). That is, a co-tensor is of power $n$ when, upon local scale change, it satisfies:

$$l' \to \lambda(x)l : \ Y'_{\mu\nu} \to \lambda^n Y_{\mu\nu} \tag{2}$$

#### 2.1.2. Dirac Co-Calculus

In the Dirac co-calculus, this results in the appearance of the "connexion" vector field $\kappa_\mu$ in the covariant derivatives of scalars, vectors, and tensors (see Table 1):

**Table 1.** Derivatives for co-tensors of power $n$.

| Co-Tensor Type | Mathematical Expression |
|---|---|
| co-scalar | $S_{*\mu} = \partial_\mu S - n\kappa_\mu S,$ |
| co-vector | $A_{\nu*\mu} = \partial_\mu A_\nu - {}^*\Gamma^\alpha_{\nu\mu} A_\alpha - n\kappa_\nu A_\mu,$ |
| co-covector | $A^\nu_{*\mu} = \partial_\mu A^\nu + {}^*\Gamma^\nu_{\mu\alpha} A^\alpha - nk^\nu A_\mu.$ |

where the usual Christoffel symbol $\Gamma^\nu_{\mu\alpha}$ is replaced by

$$^*\Gamma^\nu_{\mu\alpha} = \Gamma^\nu_{\mu\alpha} + g_{\mu\alpha}k^\nu - g^\nu_\mu \kappa_\alpha - g^\nu_\alpha \kappa_\mu. \tag{3}$$

The corresponding equation of the geodesics within the WIG was first introduced in 1973 by Dirac [9] and in the weak-field limit of Weyl gauge change redivided in 1979 by Maeder and Bouvier [10] ($u^\mu = dx^\mu/ds$ is the four-velocity):

$$u^\mu_{*\nu} = 0 \Rightarrow \frac{du^\mu}{ds} + {}^*\Gamma^\mu_{\nu\rho}u^\nu u^\rho + \kappa_\nu u^\nu u^\mu = 0. \tag{4}$$

This geodesic equation has also been derived from reparametrisation-invariant action in 1978 by Bouvier and Maeder [11]:

$$\delta\mathcal{A} = \int_{P_0}^{P_1} \delta(d\widetilde{s}) = \int \delta(\beta ds) = \int \delta\left(\beta \frac{ds}{d\tau}\right) d\tau = 0.$$

### 2.2. Consequences of Going beyond the EGR

Before we go into a specific examples, such as FLRW cosmology and weak-field limit, we would like to make few remarks. By using (3) in (4), one can see that the usual EGR equations of motion receive extra terms proportional to the four-velocity and its normalization:

$$\frac{du^\mu}{ds} + \Gamma^\mu_{\nu\rho}u^\nu u^\rho = (\kappa \cdot u)u^\mu - (u \cdot u)\kappa^\mu \tag{5}$$

In the weak-field approximation within the SIV, one assumes an isotropic and homogeneous space for the derivation of the terms beyond the usual Newtonian equations [11]. As seen from (5), the result is a velocity dependent extra term $\kappa_0\vec{v}$ with $\kappa_0 = -\dot{\lambda}/\lambda$ and $\vec{\kappa} = 0$ due to the assumption of isotropic and homogeneous space. At this point, it is important to stress that the usual normalization for the four-velocity, $u \cdot u = \pm 1$ with sign related to the signature of the metric tensor $g_{\mu\nu}$, is a special choice of $s$-parametrization—the proper-time parametrization $\tau$.

Recently, similar $\kappa_0\vec{v}$ term was derived as a consequence of non-reparametrization invariant mathematical modeling but without the need for a weak-field approximation. The effect is due to un-proper time parametrization manifested as velocity dependent fictitious acceleration [1]. In this respect, the term $\kappa_0\vec{v}$ is necessary for the restoration of the broken symmetry—the re-parametrization invariance of a process under study. To demonstrate this, one can apply an arbitrary time re-parametrization $\lambda = dt/d\tau$; then, the first term on the LHS of (5) becomes:

$$\lambda \frac{d}{dt}\left(\lambda \frac{d\vec{r}}{dt}\right) = \lambda^2 \frac{d^2\vec{r}}{dt^2} + \lambda\dot{\lambda}\frac{d\vec{r}}{dt}. \tag{6}$$

By moving the term linear in the velocity to the RHS, dividing by $\lambda^2$, and by using $\kappa(t) = -\dot{\lambda}/\lambda$, one obtains a $\kappa_0\vec{v}$-like term on the RHS. If we were to do such manipulation in the absence of $\kappa_0\vec{v}$ on the LHS of (5), then the term will be generated, while if $\tilde{\kappa}$ was present then it will be transformed $\tilde{\kappa} \to \kappa + \tilde{\kappa}$.

Furthermore, unlike in SIV, where one can justify $\lambda(t) = t_0/t$, for re-parametrization symmetry the time dependence of $\lambda(t)$ could be arbitrary. Finally, as discussed in [1], the extra term $\kappa_0\vec{v}$ is not expected to be present when the time parametrization of the process is the proper time of the system. Thus, a term of the form $\kappa\vec{v}$ can be viewed as restoration of the re-parametrization symmetry and an indication of un-proper time parametrization of a process under consideration.

In the case of the FLRW cosmology, with the assumption of homogeneity and isotropy of space, one assumes $-c^2d\tau^2 = -c^2dt^2 + a(t)^2d\Sigma^2$, where $c$ is the speed of light (to be set to 1), $\Sigma$ is a three-dimensional space of uniform curvature, and $a(t)$ is the scale factor for

the three-dimensional space. Here, $\tau$ is the proper time parametrization, presumably of the cosmological evolution, while $t$ is the coordinate time of an observer who is studying the cosmic evolution. Upon transitioning to WIG, one would have $\lambda(x)$ multiplicative factor and, in the case of $\lambda(t)$ (time dependence only), one may argue that this factor could be absorbed into $a(t)$ along with a suitable redefinition of the coordinate time $t$ into $d\tilde{t} = \lambda(t)dt$. However, this does not guarantee proper-time parametrization overall. It is therefore likely to have un-proper time parametrization for the FLRW cosmology equations, unless one makes sure that the re-parametrization symmetry is restored. This should translate into scale invariance for general $\lambda(x)$ conformal transformation.

*2.3. Scale Invariant Cosmology*

The scale invariant cosmology equations were first introduced in 1973 by Dirac [9] and then re-derived in 1977 by Canuto et al. [12]. The equations are based on the corresponding expressions of the Ricci tensor and the relevant extension of the Einstein equations.

2.3.1. The Einstein Equation for Weyl's Geometry

The conformal transformation ($g'_{\mu\nu} = \lambda^2 g_{\mu\nu}$) of the metric tensor $g_{\mu\nu}$ in the more general Weyl's framework into Einstein's framework, where the metric tensor is $g'_{\mu\nu}$, induces a simple relation between the Ricci tensor and scalar in Weyl's Integrable Geometry and the Einstein GR framework (using prime to denote Einstein GR framework objects):

$$R_{\mu\nu} = R'_{\mu\nu} - \kappa_{\mu;\nu} - \kappa_{\nu;\mu} - 2\kappa_\mu \kappa_\nu + 2g_{\mu\nu}\kappa^\alpha \kappa_\alpha - g_{\mu\nu}\kappa^\alpha_{;\alpha},$$
$$R = R' + 6\kappa^\alpha \kappa_\alpha - 6\kappa^\alpha_{;\alpha}.$$

When considering the Einstein equation along with the above expressions, one obtains:

$$R_{\mu\nu} - \frac{1}{2}g_{\mu\nu}R = -8\pi G T_{\mu\nu} - \Lambda g_{\mu\nu}, \tag{7}$$

$$R'_{\mu\nu} - \frac{1}{2}g_{\mu\nu}R' - \kappa_{\mu;\nu} - \kappa_{\nu;\mu} - 2\kappa_\mu \kappa_\nu + 2g_{\mu\nu}\kappa^\alpha_{;\alpha} - g_{\mu\nu}\kappa^\alpha \kappa_\alpha = \\ -8\pi G T_{\mu\nu} - \Lambda g_{\mu\nu}. \tag{8}$$

The relationship $\Lambda = \lambda^2 \Lambda_E$ of $\Lambda$ in WIG to the Einstein cosmological constant $\Lambda_E$ in the EGR was present in the original form of the equations to provide explicit scale invariance. This relationship makes explicit the appearance of $\Lambda_E$ as invariant scalar (in-scalar), as then one has $\Lambda g_{\mu\nu} = \lambda^2 \Lambda_E g_{\mu\nu} = \Lambda_E g'_{\mu\nu}$.

The above equations are a generalization of the original Einstein GR equation. Thus, they have an even larger class of local gauge symmetries that need to be fixed by a gauge choice. In Dirac's work, the gauge choice was based on the large numbers hypothesis. Here, we discuss a different gauge choice.

The corresponding scale-invariant FLRW based cosmology equations within the WIG framework were first introduced in 1977 by Canuto et al. [12]:

$$\frac{8\pi G\varrho}{3} = \frac{k}{a^2} + \frac{\dot{a}^2}{a^2} + 2\frac{\dot{\lambda}\,\dot{a}}{\lambda\,a} + \frac{\dot{\lambda}^2}{\lambda^2} - \frac{\Lambda_E \lambda^2}{3}, \tag{9}$$

$$-8\pi G p = \frac{k}{a^2} + 2\frac{\ddot{a}}{a} + 2\frac{\ddot{\lambda}}{\lambda} + \frac{\dot{a}^2}{a^2} + 4\frac{\dot{a}\,\dot{\lambda}}{a\,\lambda} - \frac{\dot{\lambda}^2}{\lambda^2} - \Lambda_E \lambda^2. \tag{10}$$

These equations clearly reproduce the standard FLRW equations in the limit $\lambda = const = 1$. The scaling of $\Lambda$ was recently used to revisit the Cosmological Constant Problem within quantum cosmology [13]. The conclusion of [13] is that our universe is unusually large, given that the expected mean size of all universes, where Einstein GR holds, is expected to be of a Plank scale. In the study, $\lambda = const$ was a key assumption as the universes were expected to obey the Einstein GR equations. What the expected mean size of all universes would be if the condition $\lambda = const$ is relaxed, as for a WIG-universes ensemble, remains an open question.

2.3.2. The Scale Invariant Vacuum Gauge ($T = 0$ and $R' = 0$)

The idea of the Scale Invariant Vacuum was introduced first in 2017 by Maeder [14]. It is based on the fact that, for Ricci flat ($R'_{\mu\nu} = 0$) Einstein GR vacuum ($T_{\mu\nu} = 0$), one obtains from (8) the following equation for the vacuum:

$$\kappa_{\mu;\nu} + \kappa_{\nu;\mu} + 2\kappa_\mu\kappa_\nu - 2g_{\mu\nu}\kappa^\alpha_{;\alpha} + g_{\mu\nu}\kappa^\alpha\kappa_\alpha = \Lambda\, g_{\mu\nu} \tag{11}$$

For homogeneous and isotropic WIG-space $\partial_i\lambda = 0$; therefore, only $\kappa_0 = -\dot\lambda/\lambda$ and its time derivative $\dot\kappa_0 = -\kappa_0^2$ can be non-zero. As a corollary of (11), one can derive the following set of equations [14]:

$$3\frac{\dot\lambda^2}{\lambda^2} = \Lambda, \quad \text{and} \quad 2\frac{\ddot\lambda}{\lambda} - \frac{\dot\lambda^2}{\lambda^2} = \Lambda, \tag{12}$$

$$\text{or} \quad \frac{\ddot\lambda}{\lambda} = 2\frac{\dot\lambda^2}{\lambda^2}, \quad \text{and} \quad \frac{\ddot\lambda}{\lambda} - \frac{\dot\lambda^2}{\lambda^2} = \frac{\Lambda}{3}. \tag{13}$$

One could derive these equations by using the time and space components of the equations or by looking at the relevant trace invariant along with the relationship $\dot\kappa_0 = -\kappa_0^2$. Any pair of these equations is sufficient to prove the other pair of equations.

**Theorem 1.** *Using the SIV Equations* (12) *or* (13) *with* $\Lambda = \lambda^2\Lambda_E$ *one has:*

$$\Lambda_E = 3\frac{\dot\lambda^2}{\lambda^4}, \quad \text{with} \quad \frac{d\Lambda_E}{dt} = 0. \tag{14}$$

**Corollary.** *The solution of the SIV equations is:*

$$\lambda = t_0/t, \tag{15}$$

*with* $t_0 = \sqrt{3/\Lambda_E}$ *and* $c = 1$ *for the speed of light.*

Upon the use of the SIV gauge, first in 2017 by Maeder [14], one observes that *the cosmological constant disappears* from Equations (9) and (10):

$$\frac{8\pi G\varrho}{3} = \frac{k}{a^2} + \frac{\dot a^2}{a^2} + 2\frac{\dot a\dot\lambda}{a\lambda}, \tag{16}$$

$$-8\pi G p = \frac{k}{a^2} + 2\frac{\ddot a}{a} + \frac{\dot a^2}{a^2} + 4\frac{\dot a\dot\lambda}{a\lambda}. \tag{17}$$

## 3. Comparisons and Applications

The predictions and outcomes of the SIV paradigm were confronted with observations in a series of papers by the current authors. Highlighting the main results and outcomes is the subject of current section.

### 3.1. Comparing the Scale Factor $a(t)$ within $\Lambda$CDM and SIV

Upon arriving at the SIV cosmology Equations (16) and (17), along with the gauge fixing (14), which implies $\lambda = t_0/t$ with $t_0$ indicating the current age of the universe since the Big-Bang ($a = 0$ and $t = 0$), the implications for cosmology were first discussed by Maeder [14] and later reviewed by Maeder and Gueorguiev [15]. The most important point in comparing $\Lambda$CDM and SIV cosmology models is the existence of SIV cosmology with slightly different parameters but almost the same curve for the standard scale parameter $a(t)$ when the time scale is set so that $t_0 = 1$ now [14,15]. As seen in Figure 1, the differences between the $\Lambda$CDM and SIV models declines for increasing matter densities.

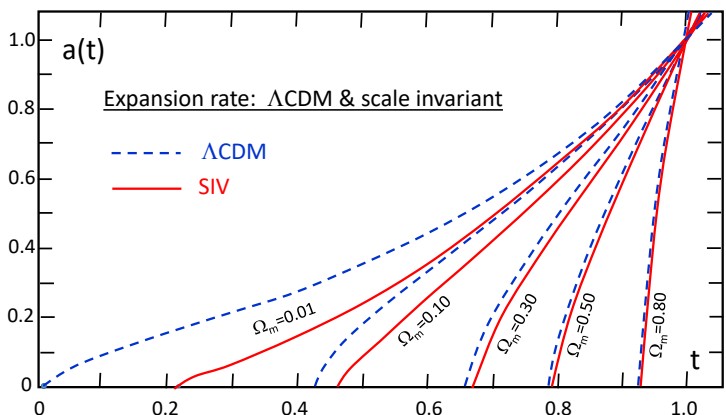

**Figure 1.** Expansion rates $a(t)$ as a function of time $t$ in the flat ($k = 0$) $\Lambda$CDM and SIV models in the matter dominated era. The curves are labeled by the values of $\Omega_{\rm m}$.

### 3.2. Application to Scale-Invariant Dynamics of Galaxies

The next important application of the scale-invariance at cosmic scales is the derivation of a universal expression for the Radial Acceleration Relation (RAR) of $g_{\rm obs}$ and $g_{\rm bar}$. That is, the relation between the observed gravitational acceleration $g_{\rm obs} = v^2/r$ and the acceleration from the baryonic matter due to the standard Newtonian gravity $g_{\rm N}$ by [4]:

$$g = g_{\rm N} + \frac{k^2}{2} + \frac{1}{2}\sqrt{4g_{\rm N}k^2 + k^4},\tag{18}$$

where $g = g_{\rm obs}$, $g_N = g_{\rm bar}$. For $g_{\rm N} \gg k^2 : g \to g_{\rm N}$ but for $g_{\rm N} \to 0 \Rightarrow g \to k^2$ is a constant.

As seen in Figure 2, MOND deviates significantly for the data on the Dwarf Spheroidals. This is well-known problem in MOND due to the need of two different interpolating functions, one in galaxies and one at cosmic scales. The SIV universal expression (18) resolves this issue naturally, with one universal parameter $k^2$ related to the gravity at large distances [4].

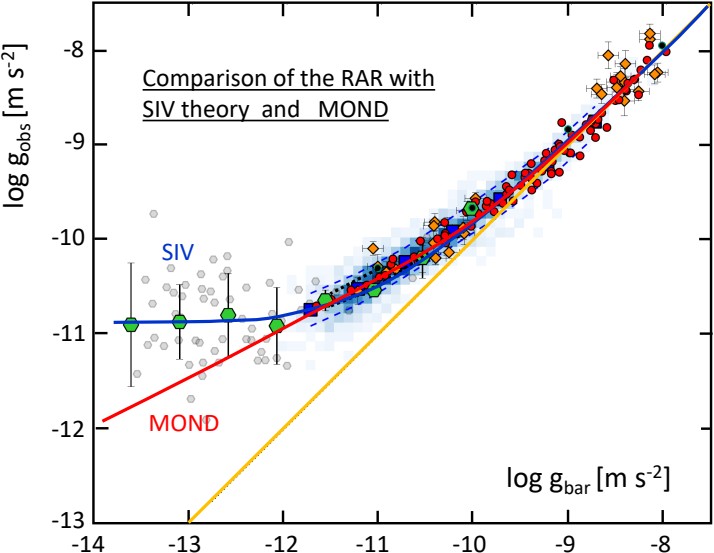

**Figure 2.** Radial Acceleration Relation (RAR) for the galaxies studied by Lelli et al. (2017). Dwarf Spheroidals as binned data (big green hexagons), along with MOND (red curve), and SIV (blue curve) model predictions. The orange curve shows the 1:1-line for $g_{\rm obs}$ and $g_{\rm bar}$. Due to to the smallness of $g_{\rm obs}$ and $g_{\rm bar}$ the application of the log function results in negative numbers; thus, the corresponding axes' values are all negative.

The expression (18) follows from the Weak Field Approximation (WFA) of the SIV upon utilization of the Dirac co-calculus in the derivation of the geodesic equation within the relevant WIG (4) (see Maeder and Gueorguiev [4] for more details, as well as the original derivation in Maeder and Bouvier [10]):

$$g_{ii} = -1, \; g_{00} \;=\; 1 + 2\Phi/c^2 \Rightarrow \Gamma^i_{00} = \frac{1}{2}\frac{\partial g_{00}}{\partial x^i} = \frac{1}{c^2}\frac{\partial \Phi}{\partial x^i},$$

$$\frac{d^2\vec{r}}{dt^2} \;=\; -\frac{GM}{r^2}\frac{\vec{r}}{r} + \kappa_0(t)\frac{d\vec{r}}{dt}. \tag{19}$$

where $i \in 1, 2, 3$, while the potential $\Phi = GM/r$ is scale invariant.

By considering the scale-invariant ratio of the correction term $\kappa_0(t)\,v$ to the usual Newtonian term in (19), one has:

$$x = \frac{\kappa_0 v r^2}{GM} = \frac{H_0}{\varsigma}\frac{v\,r^2}{GM} = \frac{H_0}{\varsigma}\frac{(r\,g_{\text{obs}})^{1/2}}{g_{\text{bar}}} \sim \frac{g_{\text{obs}} - g_{\text{bar}}}{g_{\text{bar}}}, \tag{20}$$

Then, by utilizing an explicit scale invariance for canceling the proportionality factor:

$$\left(\frac{g_{\text{obs}} - g_{\text{bar}}}{g_{\text{bar}}}\right)_2 \div \left(\frac{g_{\text{obs}} - g_{\text{bar}}}{g_{\text{bar}}}\right)_1 = \left(\frac{g_{\text{obs},2}}{g_{\text{obs},1}}\right)^{1/2}\left(\frac{g_{\text{bar},1}}{g_{\text{bar},2}}\right), \tag{21}$$

by setting $g = g_{\text{obs},2}$, $g_N = g_{\text{bar},2}$, and with $k = k_{(1)}$ all the system-1 terms, one obtains (18):

$$\frac{g}{g_N} - 1 = k_{(1)}\frac{g^{1/2}}{g_N} \Rightarrow g \;=\; g_N + \frac{k^2}{2} \pm \frac{1}{2}\sqrt{4g_N k^2 + k^4}.$$

### 3.3. Growth of the Density Fluctuations within the SIV

Another interesting result was the possibility of a fast growth of the density fluctuations within the SIV [3]. This study accordingly modifies the relevant equations such as the continuity equation, Poisson equation, and Euler equation within the SIV framework. Here, we outline the main equations and the relevant results.

By using the notation $\kappa = \kappa_0 = -\dot{\lambda}/\lambda = 1/t$, the corresponding Continuity, Poisson, and Euler equations are:

$$\frac{\partial \rho}{\partial t} + \vec{\nabla} \cdot (\rho\vec{v}) = \kappa\left[\rho + \vec{r} \cdot \vec{\nabla}\rho\right], \; \vec{\nabla}^2\Phi = \triangle\Phi = 4\pi G\varrho,$$

$$\frac{d\vec{v}}{dt} = \frac{\partial \vec{v}}{\partial t} + \left(\vec{v} \cdot \vec{\nabla}\right)\vec{v} = -\vec{\nabla}\Phi - \frac{1}{\rho}\vec{\nabla}p + \kappa\vec{v}.$$

For a density perturbation $\varrho(\vec{x}, t) = \varrho_b(t)(1 + \delta(\vec{x}, t))$ the above equations result in:

$$\dot{\delta} + \vec{\nabla} \cdot \dot{\vec{x}} = \kappa\vec{x} \cdot \vec{\nabla}\delta = n\kappa(t)\delta \quad, \quad \vec{\nabla}^2\Psi = 4\pi Ga^2\varrho_b\delta, \tag{22}$$

$$\ddot{\vec{x}} + 2H\dot{\vec{x}} + (\dot{\vec{x}} \cdot \vec{\nabla})\dot{\vec{x}} \;=\; -\frac{\vec{\nabla}\Psi}{a^2} + \kappa(t)\dot{\vec{x}}. \tag{23}$$

$$\Rightarrow \ddot{\delta} + (2H - (1+n)\kappa)\dot{\delta} \;=\; 4\pi G\varrho_b\delta + 2n\kappa(H - \kappa)\delta. \tag{24}$$

The final result (24) recovers the standard equation in the limit of $\kappa \to 0$. The simplifying assumption $\vec{x} \cdot \vec{\nabla}\delta(x) = n\delta(x)$ in (22) introduces the parameter $n$ that measures the perturbation type (shape). For example, a spherically symmetric perturbation would have $n = 2$. As seen in Figure 3, perturbations for various values of $n$ are resulting in faster growth of the density fluctuations within the SIV than in the Einstein–de Sitter model, even at relatively law matter densities. Furthermore, the overall slope is independent of the choice of recombination epoch $z_{\text{rec}}$. The behavior for different $\Omega_m$ is also interesting, and is shown and discussed in detail by Maeder and Gueorguiev [3].

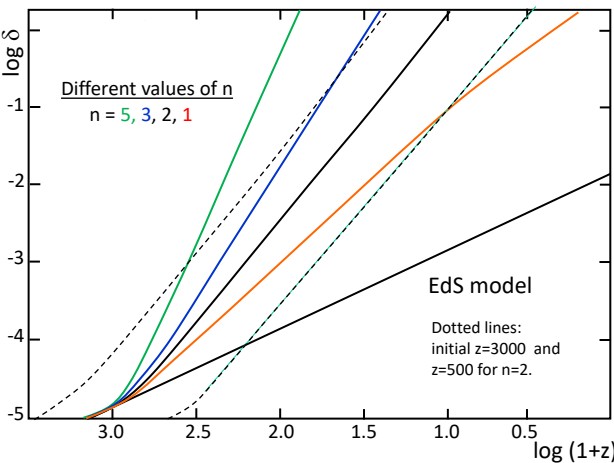

**Figure 3.** The growth of density fluctuations for different values of parameter $n$ (the gradient of the density distribution in the nascent cluster), for an initial value $\delta = 10^{-5}$ at $z = 1376$ and $\Omega_m = 0.10$. The initial slopes are those of the EdS models. The two light broken curves show models with initial $(z+1) = 3000$ and 500, with same $\Omega_m = 0.10$ and $n = 2$. These dashed lines are to be compared to the black continuous line of the $n = 2$ model. All the three lines for $n = 2$ are very similar and nearly parallel. Due to to the smallness of $\delta$ the application of the log function results in negative numbers; thus, the corresponding vertical axes values are all negative.

### 3.4. SIV and the Inflation of the Early Universe

The latest result within the SIV paradigm is the presence of inflation stage at the very early universe $t \approx 0$ with a natural exit from inflation in a later time $t_{\text{exit}}$ with value related to the parameters of the inflationary potential [2]. The main steps towards these results are outlined below.

If we go back to the general scale-invariant cosmology Equation (9), we can identify a vacuum energy density expression that relates the Einstein cosmological constant with the energy density as expressed in terms of $\kappa = -\dot{\lambda}/\lambda$ by using the SIV result (14). The corresponding vacuum energy density $\rho$, with $C = 3/(4\pi G)$, is then:

$$\rho = \frac{\Lambda}{8\pi G} = \lambda^2 \rho' = \lambda^2 \frac{\Lambda_E}{8\pi G} = \frac{3}{8\pi G}\frac{\dot{\lambda}^2}{\lambda^2} = \frac{C}{2}\dot{\psi}^2.$$

This provides a natural connection to inflation within the SIV via $\dot{\psi} = -\dot{\lambda}/\lambda$ or $\psi \propto \ln(t)$. The equations for the energy density, pressure, and Weinberg's condition for inflation within the standard model for inflation [16–19] are:

$$\left.\begin{array}{c}\rho\\p\end{array}\right\} = \frac{1}{2}\dot{\varphi}^2 \pm V(\varphi), \ |\dot{H}_{\text{infl}}| \ll H_{\text{infl}}^2. \tag{25}$$

If we make the identification between the standard model for inflation above with the fields present within the SIV (using $C = 3/(4\pi G)$):

$$\dot{\psi} = -\dot{\lambda}/\lambda, \quad \varphi \leftrightarrow \sqrt{C}\,\psi, \quad V \leftrightarrow CU(\psi), \quad U(\psi) = g\,e^{\mu\,\psi}. \tag{26}$$

Here, $U(\psi)$ is the inflation potential with strength $g$ and field "coupling" $\mu$. One can evaluate the Weinberg's condition for inflation (25) within the SIV framework [2], and the result is:

$$\frac{|\dot{H}_{\text{infl}}|}{H_{\text{infl}}^2} = \frac{3\,(\mu+1)}{g\,(\mu+2)}\,t^{-\mu-2} \ll 1 \textit{ for } \mu < -2, \textit{ and } t \ll t_0 = 1. \tag{27}$$

From this expression, one can see that there is a graceful exit from inflation at the later time:

$$t_{\text{exit}} \approx \sqrt[n]{\frac{n\,g}{3\,(n+1)}} \qquad \text{with} \qquad n = -\mu - 2 > 0, \tag{28}$$

when the Weinberg's condition for inflation (25) is not satisfied anymore.

The derivation of the Equation (27) starts with the use of the scale invariant energy conservation equation within SIV [2,14]:

$$\frac{d(\varrho a^3)}{da} + 3\,pa^2 + (\varrho + 3\,p)\frac{a^3}{\lambda}\frac{d\lambda}{da} = 0, \tag{29}$$

which has the following equivalent form:

$$\dot{\varrho} + 3\,\frac{\dot{a}}{a}\,(\varrho + p) + \frac{\dot{\lambda}}{\lambda}\,(\varrho + 3p) = 0. \tag{30}$$

By substituting the expressions for $\rho$ and $p$ from (25) along with the SIV identification (26) within the SIV expression (30), one obtains modified form of the Klein–Gordon equation, which could be non-linear when using non-linear potential $U(\psi)$ as in (26):

$$\ddot{\psi} + U' + 3H_{\text{infl}}\,\dot{\psi} - 2\,(\dot{\psi}^2 - U) = 0. \tag{31}$$

The above Equation (31) can be used to evaluate the time derivative of the Hubble parameter. The process is utilizing (14); that is, $\lambda = t_0/t$, $\dot{\psi} = -\dot{\lambda}/\lambda = 1/t \Rightarrow \ddot{\psi} = -\dot{\psi}^2$ along with $\psi = \ln(t) + const$ and $U(\psi) = g\,e^{\mu\,\psi} = gt^\mu$ when the normalization of the field $\psi$ is chosen so that $\psi(t_0) = \ln(t_0) = 0$ for $t_0 = 1$ at the current epoch. The final result is:

$$H_{\text{infl}} = \dot{\psi} - \frac{2\,U}{3\,\dot{\psi}} - \frac{U'}{3\,\dot{\psi}} \;\; = \;\; \frac{1}{t} - \frac{(2+\mu)\,g}{3}\,t^{\mu+1}, \tag{32}$$

$$\dot{H}_{\text{infl}} = -\dot{\psi}^2 - \frac{2U}{3} - U' - \frac{U''}{3} \;\; = \;\; -\frac{1}{t^2} - \frac{(\mu+2)(\mu+1)\,g}{3}\,t^\mu. \tag{33}$$

For $\mu < -2$ the $t^\mu$ terms above are dominant; thus, the critical ratio (25) for the occurrence of inflation near $t \approx 0$ is then:

$$\frac{|\,\dot{H}_{\text{infl}}\,|}{H^2_{\text{infl}}} \;\; = \;\; \frac{3\,(\mu+1)}{g\,(\mu+2)}\,t^{-\mu-2}.$$

## 4. Conclusions and Outlook

From the highlighted results in the previous section on various comparisons and potential applications, we see that the *SIV cosmology is a viable alternative to* ΛCDM. In particular, within the SIV gauge (16) *the cosmological constant disappears*. There are diminishing differences in the values of the scale factor $a(t)$ within ΛCDM and SIV at higher densities as emphasized in the discussion of (Figure 1) [14,15]. Furthermore, the SIV also shows consistency for $H_0$ and the age of the universe, and the m-z diagram is well satisfied—see Maeder and Gueorguiev [15] for details.

Furthermore, *the SIV provides the correct RAR for dwarf spheroidals* (Figure 2) while MOND is failing, and dark matter cannot account for the phenomenon [4]. Therefore, it seems that *within the SIV, dark matter is not needed to seed the growth of structure* in the universe, as there is a fast enough growth of the density fluctuations as seen in (Figure 3) and discussed in more detail by Maeder and Gueorguiev [3].

In our latest studies on the inflation within the SIV cosmology [2], we have identified a connection of the scale factor $\lambda$, and its rate of change, with the inflation field $\psi \to \varphi$, $\dot{\psi} = -\dot{\lambda}/\lambda$ (26). As seen from (27), *inflation of the very-very early universe ($t \approx 0$) is natural, and SIV predicts a graceful exit from inflation* (see (28))!

Some of the obvious future research directions are related to the primordial nucleosynthesis, where preliminary results show a satisfactory comparison between SIV and observations [20]. The recent success of the R-MOND in the description of the CMB [21], after the initial hope and concerns [22], is very stimulating and demands testing SIV cosmology against the MOND and ΛCDM successes in the description of the CMB, the Baryonic Acoustic Oscillations, etc.

Another important direction is the need to understand the physical meaning and interpretation of the conformal factor $\lambda$. As we pointed out in Section 1.2, a general conformal factor $\lambda(x)$ seems to be linked to Jordan–Brans–Dicke scalar-tensor theory that leads to a varying Newton's constant G, which has not been detected to date. Furthermore, a spacial dependence of $\lambda(x)$ opens the door to local field excitations that should manifest as some type of fundamental scalar particles. The Higgs boson is such a particle, but a connection to Jordan–Brans–Dicke scalar-tensor theory seems a far fetched idea. On the other hand, the assumption of isotropy and homogeneity of space forces $\lambda(t)$ to depend only on time, which is not in any sense similar to the usual fundamental fields we are familiar with.

In this respect, other less obvious research directions are related to the exploration of SIV within the solar system due to the high-accuracy data available, or exploring further and in more detail the possible connection of SIV with the re-parametrization invariance. For example, it is already known by Gueorguiev and Maeder [1] that un-proper time parametrization can lead to a SIV-like equation of motion (5) and the relevant weak-field version (19).

**Author Contributions:** A.M. is the lead author and researcher on the topics discussed by the paper. He has analyzed the relevant data and has produced and published previously many of the graphs shown in the Figures. V.G.G. has verified the mathematical correctness of the equations and the relevant physics conclusions. He has independently validated the trends depicted in the graphs that have been part of the previously coauthored papers with A.M. Both co-authors have been actively involved in the writing of the paper and its draft versions. All authors have read and agreed to the published version of the manuscript.

**Funding:** This research received no external funding.

**Data Availability Statement:** No new data were created or analyzed in this study. Data sharing is not applicable to this article.

**Acknowledgments:** A.M. expresses his gratitude to his wife for her patience and support. V.G.G. is extremely grateful to his wife and daughters for their understanding and family support during the various stages of the research presented. This research did not receive any specific grant from funding agencies in the public, commercial, or not-for-profit sectors.

**Conflicts of Interest:** The authors declare no conflict of interest.

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
