# Peer review of "The Scale Invariant Vacuum Paradigm: Main Results and Current Progress"

_universe, doi:10.3390/universe8040213_

Round 1
Reviewer 1 Report
This manuscript aims at representing a very short summary of some of the main results of the SIV approach in cosmology, that were already published elsewhere.
General comment: already by reading the Abstract, one concludes that this work does not present an original set of novel scientific results but only briefly overviews already published results in the literature. Therefore, this work cannot be considered as an “Article”, but may only warrant publication as a “brief review”, if issues highlighted below are properly addressed.
- I do not agree with this statement “The laws of physics (formulae) change upon change of scale.. The underlined scale is closely related
to the presence of material content” since this not realised in Nature. For instance, Maxwell theory of electrodynamics is scale-invariant.
More generically, the form-invariance of equations of motion, and hence independence laws of physics on the energy scale is a deep
consequence of Renormalisation Group invariance of an underlined renormalisable field theory (that fundamental interactions obey).
Gravity is the only physical example of a non-renormalisable field theory where the author’s point can be considered valid. - This statement “Since these new particles and/or fields have evaded any laboratory detection for more than twenty years then it
seems plausible to turn to the other alternative - a modification of EGR.” seems favouring a modification of gravity theory as a plausible
way to explain dark matter/energy in the Universe. The fact that no observable modifications of EGR have been discovered in
direct measurements over almost a hundred years of testing can be argued against, not in favour, this “plausibility”. One can only say
that both ways — through modified gravities and through “other-fields” scenarios — must be considered, but on the same footing,
as possible scenarios until an observation disfavours one of those. There is no observable hint to argue in favour of one possibility,
or the other, yet. - Regarding the physical meaning of the scalar field \lambda, how does its presence affect the diffeomorphism invariance of GR?
In the FLRW universe, any re-scaling of the metric tensor can simply be absorbed into the scale factor a(t) leading to no physical
significance of \lambda(t), unless the GR action is non-trivially modified. The latter modifications, if significant, should be discussed
in the very beginning of their theoretical introduction. - If \lambda is treated as a physical field, how does it enter into the total energy-momentum tensor T_\mu\nu that must be placed in
the r.h.s. of the equations of motion for the macroscopic geometry? What is the potential for \lambda field? Can this field be reconciled
with cosmological scalar fields often discussed in Cosmology like quintessence or inflaton field driving the de-Sitter (exponential) evolution of the Universe? If the homogeneous field \lambda(t) has a similar impact on cosmological evolution of those more standard cosmological scalar fields (like Dark Matter - for fast oscillations about the minimum of the potential, and Dark Energy - for a slow-roll regime), the authors should perform a detailed qualitative and quantitative analysis of similarities and differences of their proposal with those more standard solutions.
- I do not see how Dark Matter can be explained in the framework of SIV approach. The amount of Dark Matter should be made consistent at very different cosmological time scales, from BBN epoch, through CMB, to astrophysical observations of DM distribution in both large and small structures (also in consistency to the Bullet Cluster, gravitational lensing data, BAO etc). If not a full calculation, at least a comprehensive discussion of how SIV can be reconciled with the wealth of these observational data sensitive to a fixed amount of Cold DM in the Universe throughout many cosmological epochs has to be done here. It is quite a surprising claim that DM is not needed in the SIV framework — there are many observations of cosmological DM (apart
from the growth of perturbations) that I foresee to not fit with SIV, and a much more detailed analysis of those implications is necessary to prove their claim. - It would be good to see a single solution for \lambda(t) function that fits both inflation and late-time acceleration, as well as providing the correct growth of perturbations and other relevant characteristics of the Universe, in a single plot. I am not convinced with the text provided that such a single solution even exists (i.e. that is NOT different for very distinct cosmological epochs).
Author Response
We are thankful to the reviewer for taking the time to read and review our paper. We found their criticism to help improve our conference contribution. Please find below our Answers to the original points raised by the reviewer.
1. I do not agree with this statement “The laws of physics (formulae) change upon change of scale.. The underlined scale is closely related to the presence of material content” since this not realised in Nature. For instance, Maxwell theory of electrodynamics is scale-invariant. More generically, the form-invariance of equations of motion, and hence independence laws of physics on the energy scale is a deep consequence of Renormalisation Group invariance of an underlined renormalisable field theory (that fundamental interactions obey). Gravity is the only physical example of a non-renormalisable field theory where the author’s point can be considered valid.
A: It seems that there is common view point about the “independence of the laws of physics on the [energy] scale,” however, we feel that the Renormalizable Quantum Field Theory argument is far from the context of our discussion. Nevertheless, we have added clarifications to the controversial text in section 1.1 on page 1 to clarify our point but do not wish to engage into discussion on issues about the renormalisation paradigm and its applications.
2. This statement “Since these new particles and/or fields have evaded any laboratory detection for more than twenty years then it seems plausible to turn to the other alternative - a modification of EGR.” seems favouring a modification of gravity theory as a plausible way to explain dark matter/energy in the Universe. The fact that no observable modifications of EGR have been discovered in direct measurements over almost a hundred years of testing can be argued against, not in favour, this “plausibility”. One can only say that both ways — through modified gravities and through “other-fields” scenarios — must be considered, but on the same footing, as possible scenarios until an observation disfavours one of those. There is no observable hint to argue in favour of one possibility, or the other, yet.
A: On page 2 in section 1.2, we have modified our statement and added more text to clarify our point better.
3. Regarding the physical meaning of the scalar field \lambda, how does its presence affect the diffeomorphism invariance of GR? In the FLRW universe, any re-scaling of the metric tensor can simply be absorbed into the scale factor a(t) leading to no physical significance of \lambda(t), unless the GR action is non-trivially modified. The latter modifications, if significant, should be discussed in the very beginning of their theoretical introduction.
A: As we have pointed out in the last section of our paper, the physical meaning of lambda is still an open question. On page 4, we also added a new section 2.2. Consequences of going beyond the EGR that discusses the consequences of re-scaling, in particular the FLRW and why re-scaling may require a scale invariant cosmology equations.
4. If \lambda is treated as a physical field, how does it enter into the total energy-momentum tensor T_\mu\nu that must be placed in the r.h.s. of the equations of motion for the macroscopic geometry? What is the potential for \lambda field? Can this field be reconciled with cosmological scalar fields often discussed in Cosmology like quintessence or inflaton field driving the de-Sitter (exponential) evolution of the Universe? If the homogeneous field \lambda(t) has a similar impact on cosmological evolution of those more standard cosmological scalar fields (like Dark Matter - for fast oscillations about the minimum of the potential, and Dark Energy - for a slow-roll regime), the authors should perform a detailed qualitative and quantitative analysis of similarities and differences of their proposal with those more standard solutions.
A: As we pointed out above, the relevant physical interpretation of lambda is still an open question in general. Nevertheless, as implied by (11) on page 5, lambda goes with the cosmological constant and as such does not necessarily contribute to the matter energy-momentum tensor. Furthermore, in section 3.4. SIV and the Inflation of the Early Universe, we have explored the possibility of a link between lambda, the “dark” energy density, and the inflanton field that have allowed us to show the presence of inflation in the early SIV universe, which is followed by a graceful exit at t_exit given by Eq (28) on page 9.
5. I do not see how Dark Matter can be explained in the framework of SIV approach. The amount of Dark Matter should be made consistent at very different cosmological time scales, from BBN epoch, through CMB, to astrophysical observations of DM distribution in both large and small structures (also in consistency to the Bullet Cluster, gravitational lensing data, BAO etc). If not a full calculation, at least a comprehensive discussion of how SIV can be reconciled with the wealth of these observational data sensitive to a fixed amount of Cold DM in the Universe throughout many cosmological epochs has to be done here. It is quite a surprising claim that DM is not needed in the SIV framework — there are many observations of cosmological DM (apart from the growth of perturbations) that I foresee to not fit with SIV, and a much more detailed analysis of those implications is necessary to prove their claim.
A: As pointed out in the last section we do realize that a lot more studies have to be done to understand SIV and compare its predictions to observations. At the moment our claim is that SIV seems to be viable model at least in the area of applications we have explored.
6. It would be good to see a single solution for \lambda(t) function that fits both inflation and late-time acceleration, as well as providing the correct growth of perturbations and other relevant characteristics of the Universe, in a single plot. I am not convinced with the text provided that such a single solution even exists (i.e. that is NOT different for very distinct cosmological epochs).
A: Within the SIV theory \lambda(t) has a well defined form since the big-bang until current times. It is given in the Corollary of Theorem 1 on page 6.
If the reviewer would like to obtain a more visual understanding of lambda(t) then we would suggest Fig.3 in the paper published by Prof. Maeder in The Astrophysical Journal 834, 194 (2017).
Reviewer 2 Report
The paper considers alternative non-Einsteinian theories of GR, in particular the Weyl’s original theory of scale-invariant GR and its modifications due to Dirac published in 1973. The article reads and feels like a conference paper reviewing some results obtained by the authors’ group in the last few years. I have checked the references and found everything in the current MS already published and even explained better there. It is not clear to me what is new here.
In a review paper I expect to see either new insights or new work explaining the math or physics in the previous papers. None is given here. Instead, the authors consistently keep referring to their published papers for more details. In my opinion this is not acceptable in a review journal article especially when there are no limits on size and with such a rather short MS.
For example, many abbreviations are not even spelled out. The Weyl and Einstein frames are mentioned without definitions. No detailed derivations are given, only initial and final results. And so on.
Also, the subject is very close to P. Jordan’s scalar-tensor theory of GR, another more general and more interesting non-Einsteinian GR. No mention of this very massive body of work is mentioned here. Only 17 papers appear at the references section. Clearly this is not a serious review article.
At the end the authors refer to preprints by them where more “positive results” can be found. Could not they try to submit these preprints for peer review first then write a “review” of them?
I suggest the authors actually try to put more content in this essentially conference article before resubmitting it as a journal paper.
Author Response
We are thankful to the reviewer for taking the time to read and review our paper. We found their criticism to help improve our conference contribution. Please find below our Answers marked with A: to the original points raised by the reviewer above the answers.
"The paper considers alternative non-Einsteinian theories of GR, in particular the Weyl’s original theory of scale-invariant GR and its modifications due to Dirac published in 1973. The article reads and feels like a conference paper reviewing some results obtained by the authors’ group in the last few years. I have checked the references and found everything in the current MS already published and even explained better there. It is not clear to me what is new here.
In a review paper I expect to see either new insights or new work explaining the math or physics in the previous papers. None is given here. Instead, the authors consistently keep referring to their published papers for more details. In my opinion this is not acceptable in a review journal article especially when there are no limits on size and with such a rather short MS."
A: Our original goal was and still is to provide a very short summary of some of the main results of the SIV approach in cosmology, that were already published elsewhere, and presented at the conference on Alternative Gravities and Fundamental Cosmology (ALTECOSMOFUN'21). As such, we submitted our paper to the Special Issue "Alternative Gravities and Fundamental Cosmology.” Now the paper is about 11 pages long with five more relevant references. The new text is colored in blue.
For example, many abbreviations are not even spelled out. The Weyl and Einstein frames are mentioned without definitions. No detailed derivations are given, only initial and final results. And so on.
A: We have carefully reviewed the abbreviations within the article and have made sure to spell them out as soon as possible (see lines 1 to 3 in the abstract) as well as the title text of subsections 1.2. and 2.2. In particular, to avoid confusion with coordinate systems and frames we removed the word frame in favor of framework (see page 4).
Also, the subject is very close to P. Jordan’s scalar-tensor theory of GR, another more general and more interesting non-Einsteinian GR. No mention of this very massive body of work is mentioned here. Only 17 papers appear at the references section. Clearly this is not a serious review article.
A: We have added a discussion on P. Jordan’s scalar-tensor theory of GR along with relevant references - see the text added in section 1.2 on page 2 and the new references on page 11.
At the end the authors refer to preprints by them where more “positive results” can be found. Could not they try to submit these preprints for peer review first then write a “review” of them?
A: We would like to point out that all the papers cited have been published in peer-review journals except for the preprint on the Evolution of the early Universe that is cited as the source for preliminary results on the primordial nucleosynthesis. As we have mentioned in the Conclusions and Outlook section primordial nucleosynthesis is part of our future research directions and we intend to publish the relevant results in the near future. The preprint reference is also provided as part of the conference proceedings since we aim to provide a record of our conference presentation. The added e-sources (e-Prints, DOIs, etc.) were intended to be of convenience to the electronic reader by facilitating easy-to-follow one-click references. In this version, we have removed these convenient web-links.
I suggest the authors actually try to put more content in this essentially conference article before resubmitting it as a journal paper.
A: We have added more material to the conference paper and have highlighted it in blue text color in the updated version. Now the paper is about 11 pages long with five more relevant references. The new text is colored in blue.
Reviewer 3 Report
This article is an extended variant of a report at an international conference devoted to the analysis of some important issues in cosmology - the possibility of describing the dynamics of inflation and density perturbations within the framework of the Scale Invariant Vacuum scheme (SIV), as well as the applications of the SIV to the analysis of the dynamics of galaxies, dark matter nature and dwarfs. The presented work contains the necessary elements of the SIV mathematical tools within the framework of the Weyl Integrable Geometry; it also discusses quite fully the main results of the approach, which, from the point of view of the authors, are some alternative to the standard cosmological model LambdaCDM. The publication of this work is useful for considering and discussing the mentioned alternative.
Author Response
We are thankful and grateful to the reviewer for taking the time to read and review our paper.
Round 2
Reviewer 1 Report
The authors have adequately addressed my earlier concerns and comments.
I can now recommend this manuscript for publication in Universe journal.
Reviewer 2 Report
All my comments were addressed. Thank you.